# On the Investigation of Frequency Characteristics of a Novel Inductive Debris Sensor

**DOI:** 10.3390/mi14030669

**Published:** 2023-03-17

**Authors:** Xianwei Wu, Hairui Liu, Zhi Qian, Zhenghua Qian, Dianzi Liu, Kun Li, Guoshuai Wang

**Affiliations:** 1State Key Laboratory of Mechanics and Control of Mechanical Structures, College of Aerospace Engineering, Nanjing University of Aeronautics and Astronautics, Nanjing 210016, China; 2School of Engineering, University of East Anglia, Norwich NR4 7TJ, UK; 3China Gas Turbine Establishment, Mianyang 621000, China; 4Nanjing Research Institute of Inner Mongolia North Heavy Industry Group Co., Ltd., Nanjing 211100, China

**Keywords:** online monitoring, assess machines, inductive debris sensor, frequency characteristics, debris particles detection

## Abstract

Lubricants have the ability to reduce frictions, prevent wear, convey metal debris particles and increase the efficiency of heat transfer; therefore, they have been widely used in mechanical systems. To assess the safety and reliability of the machine under operational conditions, the development of inductive debris sensors for the online monitoring of debris particles in lubricants has received more attention from researchers. To achieve a high-precision, high-efficiency sensor for accurate prediction on the degree of wear, the equivalent circuit model of the sensor coil has been established, and its equations discovering the relationship between the induced voltage and excitation frequency have been derived. Furthermore, the influence of excitation frequencies and metal debris on the magnetic flux density has been analyzed throughout the simulations to determine the sensor magnetic field. In order to identify a frequency range suitable for detecting both ferrous and non-ferrous materials with a high level of sensitivity, the analytical analysis and experiments have been conducted to investigate the frequency characteristics of the developed inductive debris sensor prototype and its improved inspection capability. Moreover, the developed inductive debris sensor with the noticeable frequency characteristics has been assessed and its theoretical model has been also validated throughout experimental tests. Results have shown that the detection sensitivity of non-ferrous debris by the developed sensor increases with the excitation frequency in the range of 50 kHz to 250 kHz, while more complex results for the detection of ferrous debris have been observed. The detection sensitivity decreases as the excitation frequency increases from 50 kHz to 300 kHz, and then increases with the excitation frequency from 300 kHz to 370 kHz. This leads to the effective selection of the excitation frequency in the process of inspection. In summary, the investigation into the frequency characteristics of the proposed novel inductive debris sensor has enabled its broad applications and also provided a theoretical basis and valuable insights into the development of inductive debris sensors with improved detection sensitivity.

## 1. Introduction

The gradual increase in the use of functional materials and structures to achieve advanced machinery and equipment has attracted great interest among researchers. In such systems, the surface wear arising from mechanical contacts under operational conditions has been used to assess their health status. If abnormal wear is not detected in the early stage, serious accidents may occur during operation. Therefore, machine condition monitoring is critical to maintain equipment health and extend their life cycle [1,2]. Research has shown that there is a direct correlation between the degree of wear and the concentration and size of metal debris in the lubricants of machinery and equipment. When the machine operated normally, the size of wear debris was in the range of 1 μm to 20 μm and the concentration was usually low [3]. However, as abnormal wear started, the size of wear debris between 50 μm and 100 μm was produced. Over time, the size and concentration of the wear debris would further increase [4]. Therefore, the degree of the wear can be judged by detecting the concentration and size of metal debris in the lubricants in the machinery and equipment. Owing to this observation, it can indicate the level of wear of mechanical components and provide the system with prognosis warnings before a failure occurs.

The traditional detection techniques are mainly categorized into two groups: offline and online inspection methods. Usually, ferrographic analysis and spectral analysis are conducted by the offline detection methods. It is noted that the ferrographic analysis has the ability to effectively detect ferrous debris particles, but cannot be applied for the detection of non-ferrous metal debris particles due to the non-magnetic property of non-ferrous metal debris [5]. Meanwhile, the spectral analysis can be used to identify the size of debris particles without the capability of composition analysis of debris particles [6,7]. Moreover, the offline inspection methods need long-term testing with high costs, and are difficult to be implemented for real-time monitoring of the equipment. In order to enable real-time monitoring of machinery and equipment under the operational conditions, several online metal debris detection methods have been developed. As different detection methods have different advantages and limitations, this has undoubtedly restrained the technology from industrial applications. For instance, although the X-ray method [8] has a high detection accuracy, it can only be deployed on the complex equipment. Additionally, the detection using capacitance methods [9,10] or resistance methods [11,12] will result in the oil deterioration, which will degrade the inspection accuracy as time goes. For the ultrasonic method [13], the measurement precision is affected by many factors, such as the viscosity of the oil, the flow rate, and mechanical vibration, leading to the challenge in practical applications. The inductive method [14,15,16,17,18,19,20], which can effectively distinguish nonferrous and ferrous metal debris, can be easily implemented in a simple structure of equipment for testing both metal and non-metal pipelines. Additionally, the sensitivity of this method does not rely on the oil quality. However, the inductive method has limitations, including low sensitivity to non-ferrous metal debris and the inability to detect debris shape. From a practical point of view, the inductive method is the most feasible and effective technique for engineering applications.

Since the induction method has many advantages, extensive research has been conducted in this field. The structure of induction method-based sensors mainly includes solenoid coils and planar coils. Although the planar coil has the high detection sensitivity, it is not suitable for engineering applications due to its small size. For example, the planar coil sensor designed in [21] was capable of detecting 50 μm ferrous debris particles and 105 μm non-ferrous debris particles for a 1.2 mm inner diameter of the oil tube. Due to its wider detection range and larger size, the solenoid coil structure has wider engineering applications. Using such structure, a three-dimensional solenoid sensor (the MetalSCAN from GasTOPS) was used to effectively detect the ferrous metal debris with a size of 100 µm and nonferrous metal debris with a size of 405 µm in a pipe with an inner diameter of 9.525 mm [22]. Talebi et al. [23] designed a sensor capable of effectively detecting 125 µm ferrous debris in pipes with an internal diameter of 4 mm and measuring the concentration of metal debris in the oil. Additionally, results obtained by the solenoid coil sensor [24] indicated that its sensitivity to ferrous and non-ferrous metal debris in the inner diameter of the pipe, which was approximate 43 mm, could be achieved with values of 70 µm (diameter) and 165 µm, respectively. Liu et al. [25] investigated the relationship between the excitation frequency of the microinductor sensors and the rate of change of the sensor inductance. The sensitivity of ferrous metal debris decreased with the increasing excitation frequency, while the sensitivity of non-ferrous metal debris increased with the excitation frequency.

To provide a theoretical basis for improving the detection sensitivity of inductive debris sensors and selecting a suitable excitation frequency for the detection of different metal particles, a novel inductive debris sensor design is proposed in this paper. Based on Kirchhoff’s principle, the equivalent circuit model of this sensor is established. Following that, the equation for bridging the induced voltage and excitation frequencies is derived to discover the most suitable excitation frequency of the inductive debris sensor for the detection with a high level of accuracy and sensitivity. Using COMSOL, a simulation model of the sensor magnetic field is also developed to analyze the influence of excitation frequencies and metal debris material on the magnetic flux density. Finally, to demonstrate the correctness of the developed theoretical model, the inductive debris sensor is fabricated and assessed for the detections of ferrous or non-ferrous metal particles by comparing the analytical results.

## 2. The inductive Debris Sensor

### 2.1. Working Principle of the Proposed Inductive Sensor

The proposed inductive sensor’s operating principle [26] is briefly described in Figure 1 (the red coil is the excitation coil, and the blue coil is the induction coil). An AC signal is passed through the excitation coil, and the two cases can be defined as follows: When no metal particles enter the sensor, the resulting magnetic field is shown in Figure 1a. If the sensor detects ferrous metal debris, there are two factors influencing the original magnetic field, including the magnetic flux and the eddy current shown in Figure 1b. First, the magnetic flux increases due to the higher permeability of the ferrous metal debris. Second, a magnetic field whose direction is opposite to the original magnetic field will be generated by the eddy current inside the ferrous metal debris, leading to the decrease in the total magnetic flux. Due to the smaller eddy current, the increase in the magnetic flux dominates in the frequency range of low frequencies. Therefore, a positive voltage pulse will be generated when the ferrous metal debris flows through the sensor. On the contrary, the magnetic flux of the larger eddy current decreases at high frequencies. Thus, a negative voltage pulse will be generated when ferrous debris flows through the sensor. When non-ferrous metal debris enter the sensor, it is mainly the eddy currents influencing the original magnetic field, as shown in Figure 1c.

### 2.2. Sensor Model Simplification

The equivalent circuit of the proposed sensor device is shown in Figure 2a. The resistances and inductances of the excitation and sensing coils are $R_{0}$ and $L_{0}$ as well as R and $L_{2}$, respectively. The AC voltage $\dot{U}$ is loaded on the excitation coil and the current is denoted as ${\dot{I}}_{0}$. In the inspection process, the voltage output from the sensing coil is expressed as ${\dot{u}}_{1}$ if no metal debris pass through the sensor. Thus, the output voltage ${\dot{u}}_{1}$ can be formulated as follows:(1)$${\dot{u}}_{1}=-j\omega M_{0}{\dot{I}}_{0}$$
(2)ω=2πf
where ω is the angular frequency of the excitation signal; f is the frequency of the excitation signal; and $M_{0}$ is the mutual inductance coefficient between the excitation coil and the sensing coil.

The equivalent circuit of the sensor coil for detecting ferrous metal debris is shown in Figure 2b. When the voltage $\dot{U}$ applied to the excitation coil is constant, the inductance $L_{0}$ and resistance $R_{0}$ of the excitation coil are constant. Thus, the current ${\dot{I}}_{0}$ flowing through the excitation coil is also constant. Therefore, the eddy current in ferrous metal debris is considered a short-circuited coil. The resistance of the short-circuited coil is denoted as $R_{1}$ with the inductance $L_{1}$ and the eddy current ${\dot{I}}_{1}$. The mutual inductance coefficient between the excitation coil and ferrous metal debris is expressed as $M_{1}$, and the mutual inductance coefficient between ferrous metal debris and the sensing coil is represented as $M_{2}$.

As mentioned earlier, there are two factors that cause changes in the output voltage of the sensor. One is the change in magnetic permeability. It only depends on the volume of the metal debris, the magnetic permeability of metal debris and the speed of the passage. Therefore, the change in output voltage caused by the change in permeability is a constant value, given the volume, the properties of metal debris and the speed of passage are determined. Thus, the change in output voltage can be expressed as follows:(3)$$\dot{u}=\frac{\Delta \varphi }{\Delta t}$$

As the ferrous metal debris pass through the inductive sensor, the voltage generated by the change in magnetic permeability is defined by Equation (4)
(4)$${\dot{u}}_{21}=\frac{\Delta \varphi_{1}}{\Delta t}$$
where $\Delta \varphi_{1}$ is the change in magnetic flux caused by the change in permeability when the ferrous metal debris pass through the inductive sensor.

According to the equivalent circuit shown in Figure 2b and Kirchhoff’s law, one arrives at the following:(5)$$\left\{ \begin{array}{l} R_{0}{\dot{I}}_{0}+j\omega L_{0}{\dot{I}}_{0}-j\omega M_{1}{\dot{I}}_{1}=\dot{U}\\ -j\omega M_{1}{\dot{I}}_{0}+R_{1}{\dot{I}}_{1}+j\omega L_{1}{\dot{I}}_{1}=0 \end{array}\right.$$
where ω=2πf and f is the frequency of the excitation signal.

The eddy current in the ferrous metal debris can be determined by Equation (5), where one arrives at the following:(6)$${\dot{I}}_{1}=\frac{\dot{U}\cdot j\omega M_{1}}{\left(R_{0}+j\omega L_{0}\right)\left(R_{1}+j\omega L_{1}\right)+\omega^{2}M_{1}^{2}}$$

The eddy current in the ferrous metal debris interacts with the induction coil to produce the induced voltage, which is expressed as the following:(7)$${\dot{u}}_{22}=-j\omega M_{2}{\dot{I}}_{1}$$

Substituting Equation (6) into Equation (7), one has the following:(8)$${\dot{u}}_{22}=\frac{\dot{U}\cdot M_{1}M_{2}}{\frac{R_{0}R_{1}}{\omega^{2}}+j\omega \frac{R_{0}L_{1}+R_{1}L_{0}}{\omega^{2}}-L_{0}L_{1}}=\frac{\dot{U}\cdot M_{1}M_{2}}{\frac{R_{0}R_{1}}{4\pi^{2}\cdot f^{2}}+j\omega \frac{R_{0}L_{1}+R_{1}L_{0}}{4\pi^{2}\cdot f^{2}}-L_{0}L_{1}}$$

As the change of flux produces a voltage, which is opposite to what is produced by the eddy current effect, the output voltage can be formulated as the following:(9)$$\Delta {\dot{u}}_{1}={\dot{u}}_{2}-{\dot{u}}_{1}={\dot{u}}_{21}-{\dot{u}}_{22}$$

Substituting Equations (4) and (8) into Equation (9), one has the following:(10)$$\Delta {\dot{u}}_{1}=\frac{\Delta \varphi_{1}}{\Delta t}-\frac{\dot{U}\cdot M_{1}M_{2}}{\frac{R_{0}R_{1}}{4\pi^{2}\cdot f^{2}}+j\omega \frac{R_{0}L_{1}+R_{1}L_{0}}{4\pi^{2}\cdot f^{2}}-L_{0}L_{1}}$$

When the sensor detects ferrous metal particles, it is worth noting that in Equation (10), the output voltage of the sensor at the low excitation frequency is positive. As the excitation frequency increases, the output voltage of the sensor decreases until the sensor outputs a negative voltage.

Additionally, in the case that the magnetic permeability of the non-ferrous metal particles is close to that of air, the voltage generated by the change in magnetic permeability due to non-ferrous metal particles passing through the inductive sensor can be formulated as the following:(11)$${\dot{u}}_{31}=\frac{\Delta \varphi_{2}}{\Delta t}\approx 0$$

Furthermore, the eddy currents in non-ferrous metal particles are considered an equivalent value in the state of short-circuit coils. Here, the resistance of the short-circuit coil is defined as $R_{2}$ with the inductance of $L_{3}$ and the eddy current of ${\dot{I}}_{2}$. The mutual inductance coefficient between the excitation coil and the non-ferrous metal particles is expressed as $M_{3}$. According to the equivalent circuit in Figure 2c and the Kirchhoff’s law, one has the following:(12)$$\left\{ \begin{array}{l} R_{0}{\dot{I}}_{0}+j\omega L_{0}{\dot{I}}_{0}-j\omega M_{3}{\dot{I}}_{2}=\dot{U}\\ -j\omega M_{3}{\dot{I}}_{0}+R_{2}{\dot{I}}_{2}+j\omega L_{3}{\dot{I}}_{2}=0 \end{array}\right.$$

Solving Equation (12), the eddy current in the non-ferrous metal debris can be formulated as the following:(13)$${\dot{I}}_{2}=\frac{\dot{U}\cdot j\omega M_{3}}{\left(R_{0}+j\omega L_{0}\right)\left(R_{2}+j\omega L_{3}\right)+\omega^{2}M_{3}^{2}}$$

The induced voltage that is produced by the interaction between the eddy current and the induction coil is defined as the following:(14)$${\dot{u}}_{32}=-j\omega M_{4}{\dot{I}}_{2}$$
where $M_{4}$ represents the mutual inductance coefficient between the non-ferrous metal particles and the sensing coil.

Substituting Equation (13) into Equation (14), one has the following:(15)$${\dot{u}}_{32}=\frac{\dot{U}\cdot M_{3}M_{4}}{\frac{R_{0}R_{2}}{4\pi^{2}\cdot f^{2}}+j\omega \frac{R_{0}L_{3}+R_{2}L_{0}}{4\pi^{2}\cdot f^{2}}-L_{0}L_{3}}$$

Similarly, the output voltage can be formulated as the following:(16)$$\Delta {\dot{u}}_{2}={\dot{u}}_{3}-{\dot{u}}_{1}={\dot{u}}_{31}-{\dot{u}}_{32}$$

Substituting Equation (11) into Equation (16), one has the following:(17)$$\Delta {\dot{u}}_{2}=-\frac{\dot{U}\cdot M_{3}M_{4}}{\frac{R_{0}R_{2}}{4\pi^{2}\cdot f^{2}}+j\omega \frac{R_{0}L_{3}+R_{2}L_{0}}{4\pi^{2}\cdot f^{2}}-L_{0}L_{3}}$$

Using Equation (17), the output of the sensor is a negative voltage when the sensor detects non-ferrous metal particles. As the excitation frequency increases, the amplitude of the output voltage increases.

## 3. Simulation Analysis of the Sensor Magnetic Field

According to the detection principle of the proposed sensor, the output signal of the induction coil is generated by the perturbation of the magnetic field as the metal debris passes through the sensor. Therefore, it is necessary to analyze the influence of the metal debris on the perturbation of the original magnetic field. As shown in Figure 3, a numerical simulation model is established, the red coil is the excitation coil and the blue coil is the induction coil, and the coordinate is defined as follows: the center of the coil is the origin; and the axial and radial directions of the coil are the *z*-axis and *x*-axis, respectively. Additionally, the metal debris is located at the center of the coil. Numerical simulations are carried out using COMSOL to analyze the perturbation of the magnetic field at different frequencies as the metal debris passes through the sensor. The model parameters are shown in Table 1 and the typical ferrous and non-ferrous metal debris, and their main electromagnetic parameters are provided in Table 2.

### 3.1. The Influence of Ferrous Metal Debris on the Magnetic Flux Density

Numerical simulations of 200 µm diameter iron particles shown in Figure 4 are conducted to demonstrate effects of ferrous debris on the distribution of the magnetic flux density along the *z*-axis over different frequencies at time t0 = T/2 (T is the period of the excitation signal). It can be observed that the magnitude of the magnetic flux density inside the iron particles is smaller than that of the background flux density due to the eddy current effect of metal debris. As iron particles have the property of a larger relative magnetic permeability, the magnetic flux density increases rapidly when approaching the surface of iron particles. Furthermore, the eddy current among the iron particles increases with the frequency, while the magnetic flux density near the surface of the iron particles decreases, leading to the weak influence of iron particles on the original magnetic field.

### 3.2. The Influence of Non-Ferrous Metal Debris on Magnetic Flux Density

To investigate the influence of non-ferrous metal debris on magnetic flux density, 800 μm copper particles as non-ferrous metal debris are numerically simulated in Figure 5 to illustrate the distribution of magnetic flux density along the z-axis at different frequencies at time t0 = T/2. Similarly, the flux density measured inside the copper particles is smaller than the background flux density due to the eddy current effect. As the frequency increases, the magnitude of the eddy current measured inside the copper particles increases and the magnetic flux density inside the copper particles decreases. Therefore, the effect of copper particles on the original magnetic field is increased.

## 4. Experimental Results and Discussion

### 4.1. Experimental Setup

The experimental sensor is designed with two excitation coils (E1 and E2) and two induction coils (S1 and S2), shown in Figure 6. In the experiment, the detected metal debris particles signal is a complete sinusoidal function. First, the sensing coil is fabricated by winding 0.1 mm diameter enameled wire on a skeleton with an inner diameter of 8 mm and a thickness of 1 mm, with a total of 400 turns. Similarly, the excitation coil is produced using 0.2 mm diameter enameled wire around the outside of the sensing coil (300 turns), and the skeleton is made of the epoxy resin material. As the magnetic permeability of the epoxy resin is close to that of air, the epoxy resin has little effect on the magnetic field. To facilitate the control of the frequency of the excitation signal, the signal collected by the sensor is processed by a differential amplifier circuit shown in Figure 7, and the output is displayed on a computer.

In order to simulate the passage of metal particles in the lubricant through the sensor, the metal particles are fixed on a nylon rope conveying through the sensor shown in Figure 8a. The nylon rope is then supported by three pulleys, one of which is driven by a motor that controls the speed of the metal particles passing through the sensor. It is noted that the magnetic permeability of the nylon rope is close to that of air and its effect on the magnetic field can be ignored. The entire experimental platform is shown in Figure 8b.

### 4.2. Experimental Results of Ferrous Metal Particles

The 245 μm and 480 μm iron particles are chosen as ferrous metal debris in the experiments, and their microphotograph are shown in Figure 9. The iron particle passes through the sensor at a speed of 0.4 m/s. A sinusoidal AC signal with a amplitude of ±10 V is generated, and its excitation frequency ranges from 50 kHz to 370 kHz. The output voltage waveform of 480 μm iron particles at different excitation frequencies is shown in Figure 10. It can be seen that the output voltage is positive at lower excitation frequencies and negative at higher excitation frequencies. Figure 11 shows the relationship between output voltages and excitation frequencies. Experimental results show that at lower excitation frequencies the increase in the flux dominates, and the output voltage amplitude is positive. Additionally, the eddy current increases with the excitation frequency, while the output voltage amplitude becomes smaller when the frequency increases. When the excitation frequency is greater than 300 kHz, the eddy currents continue to increase. Therefore, the eddy currents in the metal particles play a dominant role, leading to a reverse increase in the output voltage amplitude.

### 4.3. Experimental Results of Non-Ferrous Metal Particles

In this section, 800 μm copper particles are chosen as non-ferrous metal particles for the experiments. The microphotograph of copper particles is shown in Figure 12. The parameters used for the input signals are the same as in Section 4.2, but an excitation frequency in the range of 50 kHz to 250 kHz is used. The output voltage waveform of copper particles at different excitation frequencies is shown in Figure 13. It can be seen that the output voltage of copper particles at different excitation frequencies is negative. The relationship between output voltages and frequencies is demonstrated in Figure 14. Experimental results show that when the excitation frequency is higher than 50 kHz, the output voltage amplitude increases with the increase in the frequency. As the eddy current has a dominant effect on the magnetic flux density, both the non-ferrous metal eddy current and the output voltage are increased. This observation is generally consistent with the results of the theoretical model.

## 5. Conclusions

The inductive debris sensor is suitable for the online monitoring of various mechanical equipment, such as aero-engines, gas turbines, and wind turbines, as it has the capablity to accurately indicate the degree of wear in machinery components and provide a prognosis warning for the system before any fault occurs. Therefore, its prospects for broad applications can be anticipated in various engineering sectors. In this paper, the frequency characteristics of a novel inductive debris sensor have been investigated based on the equivalent circuit model of the sensor coil. Stemming from the established model, the relationship between induced voltages and excitation frequencies has been mathematically represented. To analyze the influence of excitation frequency and metal debris material on magnetic flux density, numerical simulations of the sensor magnetic field have been performed. Then, experimental tests of the fabricated inductive debris sensor prototype have been conducted to validate its correctness and effectiveness by comparison of the results obtained from the theoretical model. The output voltage of the ferrous particles has changed from positive to negative as the excitation frequency has increased from 50 kHz to 370 kHz, while the voltage amplitude for the non-ferrous particles increases with the frequency in the range of 50 kHz to 250 kHz. It should be noted that selecting the appropriate excitation frequency enables high sensitivity in detecting both ferrous and non-ferrous metal particles, and distinguishes between them based on their respective positive and negative output voltages. Furthermore, leveraging the excitation frequency, the sensor can be specially designed to effectively detect the wear of the object, e.g., ferromagnetic metal particles or non-ferromagnetic metal particles. Research studies on frequency characteristics of the proposed inductive debris sensor provide a useful insight into the development of the advanced inductive debris sensors with robust characteristics, such as excitation frequency and the sensitivity.

## Figures and Tables

**Figure 1 micromachines-14-00669-f001:**
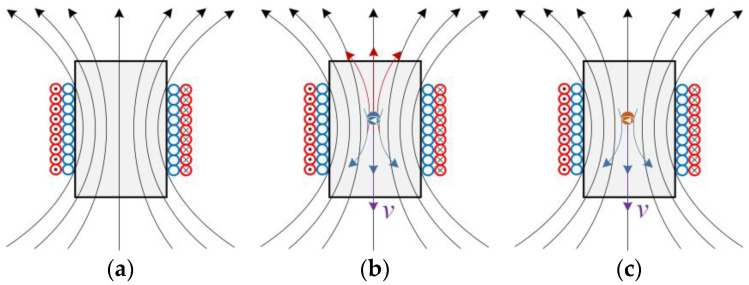
Distribution of magnetic field in the designed sensor: (**a**) no metal debris passes through; (**b**) when ferrous metal debris enters the sensor; (**c**) when non-ferrous metal debris enters the sensor.

**Figure 2 micromachines-14-00669-f002:**
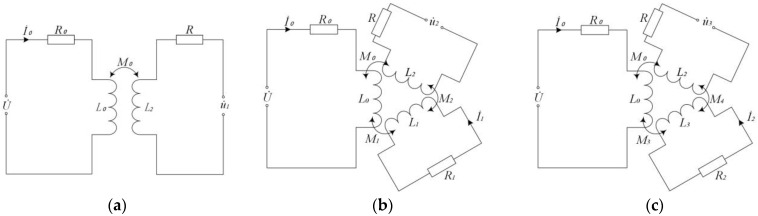
The equivalent circuit diagram: (**a**) no debris; (**b**) the test of ferrous debris; (**c**) the test of non-ferrous debris.

**Figure 3 micromachines-14-00669-f003:**
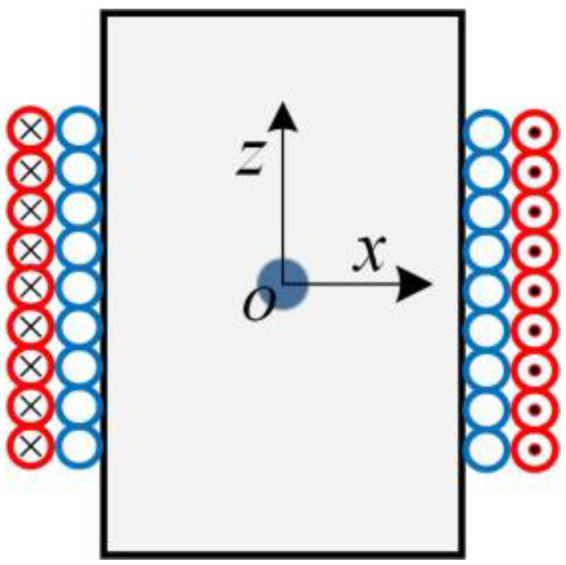
Simplified model of the coil.

**Figure 4 micromachines-14-00669-f004:**
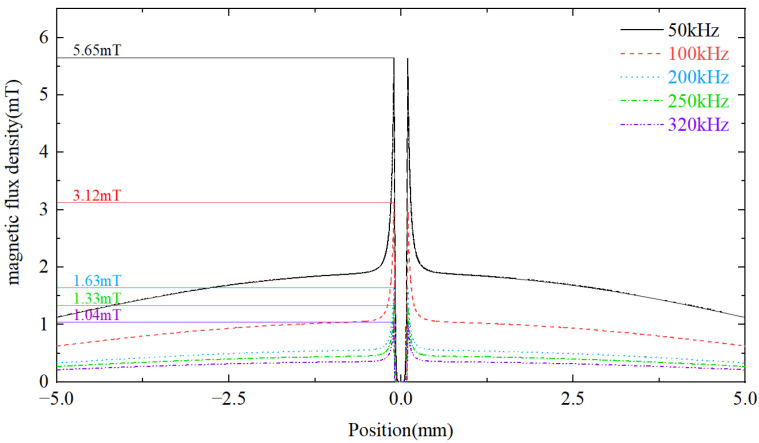
Distribution of the magnetic flux density along z-axis under the excitation of different frequencies.

**Figure 5 micromachines-14-00669-f005:**
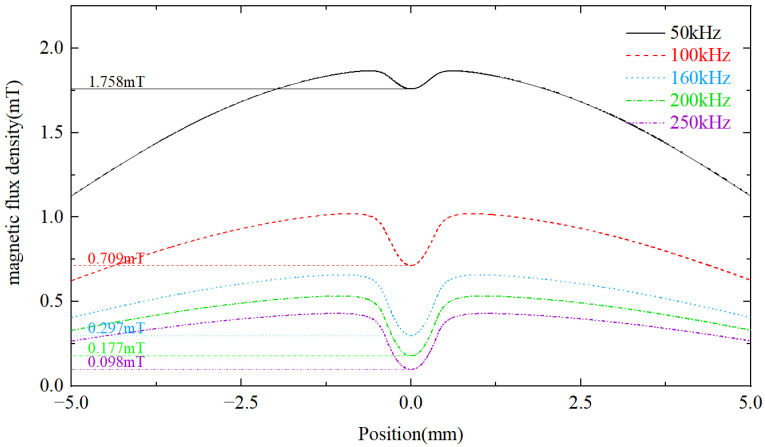
Distribution of the magnetic flux density along z-axis for non-ferrous metal debris.

**Figure 6 micromachines-14-00669-f006:**
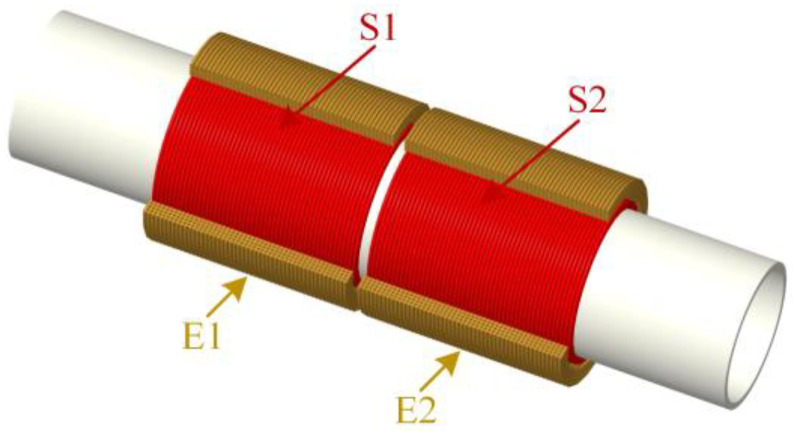
The sensor structure model: E1 and E2 are excitation coils; S1 and S2 are sensing coils.

**Figure 7 micromachines-14-00669-f007:**
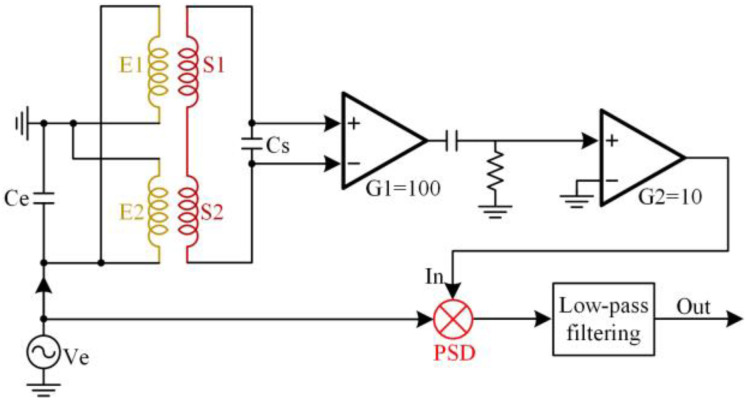
The sensor signal process.

**Figure 8 micromachines-14-00669-f008:**
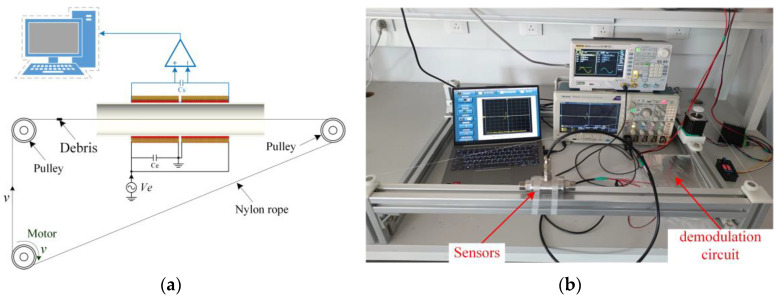
(**a**) Schematic diagram of the whole inspection system; (**b**) the experimental test platform.

**Figure 9 micromachines-14-00669-f009:**
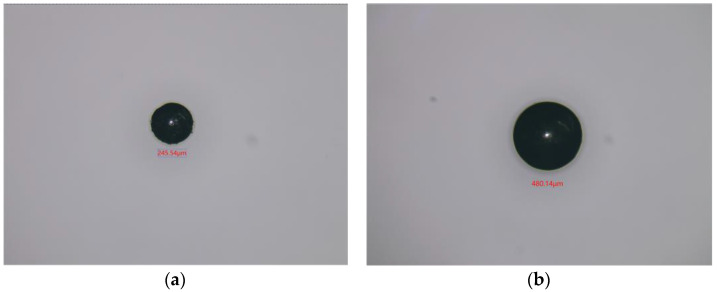
The microphotograph of iron particles. (**a**) 245 μm; (**b**) 480 μm.

**Figure 10 micromachines-14-00669-f010:**
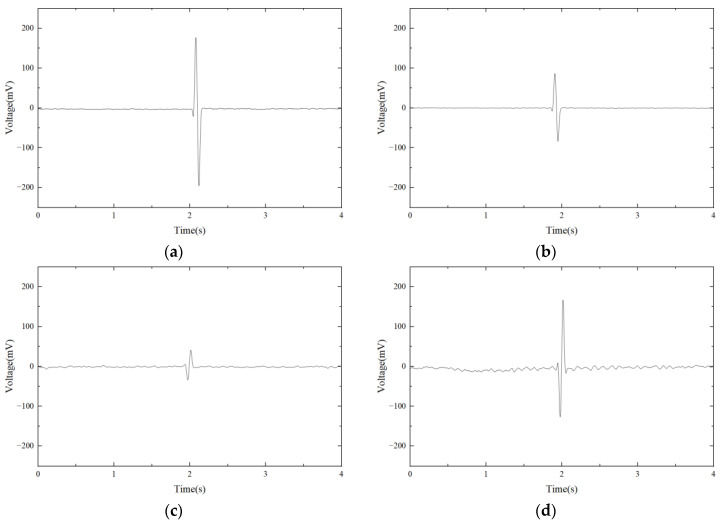
The output voltage waveform of 480 μm iron particles at different excitation frequencies. (**a**) 80 kHz; (**b**) 160 kHz; (**c**) 310 kHz; (**d**) 360 kHz.

**Figure 11 micromachines-14-00669-f011:**
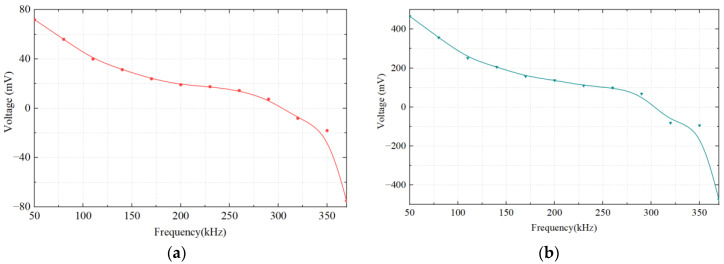
The relationship between output voltage and excitation frequency of ferrous metal particles. (**a**) 245 μm; (**b**) 480 μm.

**Figure 12 micromachines-14-00669-f012:**
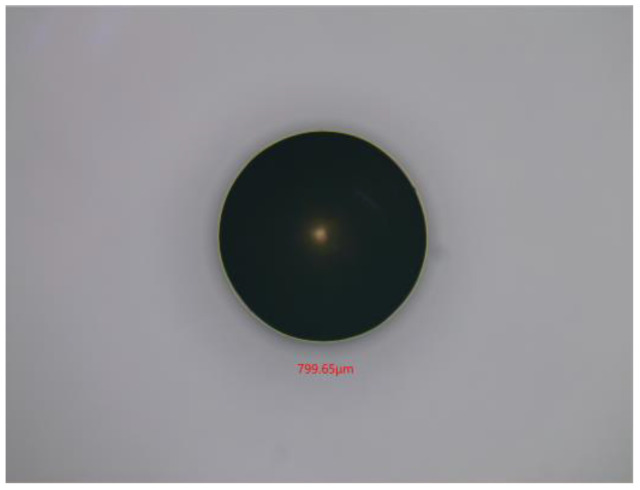
The microphotograph of 800 µm copper particles.

**Figure 13 micromachines-14-00669-f013:**
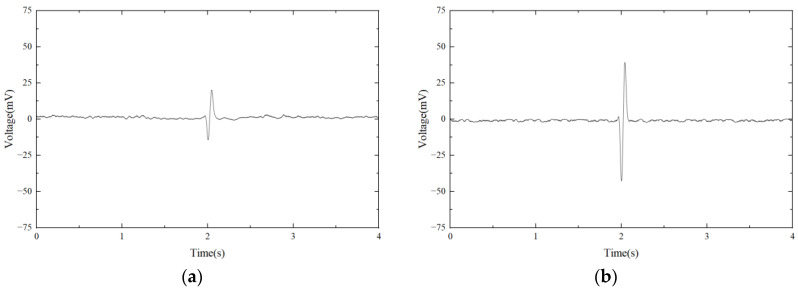
The output voltage waveform of copper particles at different excitation frequencies. (**a**) 80 kHz; (**b**) 160 kHz.

**Figure 14 micromachines-14-00669-f014:**
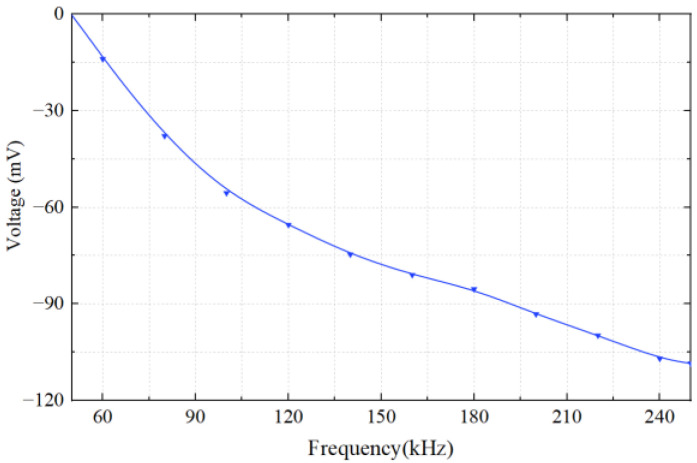
The relationship between the output voltage and excitation frequency of non-ferrous metal particles.

**Table 1 micromachines-14-00669-t001:** Numerical simulation modeling parameters.

Coil Parameters	Value	Unit
Internal diameter of induction coil	10	mm
External diameter of induction coil	10.8	mm
Wire diameter of induction coil	0.1	mm
Number of turns of induction coil	400	/
Internal diameter of excitation coil	10.8	mm
External diameter of excitation coil	13.2	mm
Wire diameter of excitation coil	0.2	mm
Number of turns of excitation coil	300	/
Coil width	10	mm
Excitation signal amplitude	10∗sin (2πft)	V

**Table 2 micromachines-14-00669-t002:** Electromagnetic parameters of metal debris.

Metal Debris Properties	Materials	Relative Permeability
ferrous metal debris	Iron	4000
non-ferrous metal debris	Copper	1

## Data Availability

The data supporting reported results can be made available via requesting the corresponding author.

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
