# Peer review of "On the Investigation of Frequency Characteristics of a Novel Inductive Debris Sensor"

_micromachines, 2023, doi:10.3390/mi14030669_

Round 1
Reviewer 1 Report
1. Authors need to highlight the novelty of work compared to the work done by others.
Suggest to include and comment on the generated voltage waveform.
2. Can authors comment of the speed of debris chosen? Is this the standard speed? 3. 3. Can authors comments on the results obtained for different speed of debris?
4. How about if there are more than 1 debris that are near to each other? Or ferrous and non ferrous metals are present at the same time? Can the prototype detect?
5. Authors tested with 245 μm and 480 μm iron and 800 μm copper particles. Did authors tried with smaller size? Other work demonstrated by other researchers were able to detect much smaller size.
Reviewer 2 Report
Reviewer’s comments:
1. In Figure 1, the symbols or patterns of ferrous debris and non-ferrous debris should be different to readable identification.
2. On page 3, the equation “ (1)” should be exhibited in the text and the both equations (1) should also be divided into equations “(1a)” and “(1b)”. On page 4, the equation (3) should be also exhibited in the text.
3. It is suggested that the comparison of detection sensitivity (S) both for non-ferrous debris and for ferrous debris should be revealed in the manuscript using a sensitivity formula to numerical sensitivity such as S= (high voltage – low voltage)/(high frequency –low frequency).
Reviewer 3 Report
In order to evaluate the safety and reliability of the machine under operating conditions and realize the accurate prediction of wear degree of high-precision and high-efficiency sensor, an equivalent circuit of sensor coil is established in this paper, and the relationship equation between induced voltage and excitation frequency is derived. The effects of ferrous and non-ferrous materials on magnetic flux density are investigated and a frequency range suitable for them is determined. In addition, the developed inductive debris sensor with obvious frequency characteristics is evaluated and its theoretical model is verified by experiments. The article is rich and smooth, with high research value, but some minor problems need to be modified:
1. Some parts of the sentence is not very coherent, please correct.
2. The article cites few references in the past five years, please add.
3. In conclusion, the prospect of this sensor is less described. Please consider whether it needs to be added.
